# Predictive value of SOFA, PCT, Lactate, qSOFA and their combinations for mortality in patients with sepsis: A systematic review and meta-analysis

Jinmei Lu[1], Zhouzhou Dong[1], Longqiang Ye[1], Yi Gao[2], Zaixing Zheng🆔[2]*

**1** Department of Critical Care Medicine, Ningbo Medical Center Li Huili Hospital, Ningbo, Zhejiang, China,
**2** Department of Cardiology, Ningbo NO. 2 Hospital, Zhejiang, China

* zzxnbeydoc@163.com

## Abstract

### Background

Sepsis is a leading cause of death, necessitating early prediction of mortality risk.

### Objective

To systematically review the predictive efficacy of the Sequential Organ Failure Assessment (SOFA), procalcitonin (PCT), lactate, quick Sequential Organ Failure Assessment (qSOFA), and lactate-adjusted qSOFA (LqSOFA) for the risk of death in patients with sepsis.

### Methods

According to PRISMA-DTA guidelines, PubMed, Embase, the Cochrane Library, and CNKI were searched (up to March 2025), and 29 studies were included (n = 41,469). A bivariate random-effects model was used to pool the sensitivity, specificity, diagnostic odds ratio, and area under the receiver operating characteristic curve (AUROC). The ΔAUROC was compared using a random-effects model based on a paired-data design. Heterogeneity was evaluated by $I^2$ (>50%) and Cochrane's Q test.

### Results

SOFA demonstrated superior predictive efficacy (AUROC = 0.819, 95% CI: 0.783–0.850; sensitivity = 0.77, 95% CI: 0.71–0.82; specificity = 0.73, 95% CI: 0.67–0.79), significantly outperforming PCT (ΔAUROC = 0.10, 95% CI: 0.04–0.16), lactate (ΔAUROC = 0.07, 95% CI: 0.03–0.11), and qSOFA (ΔAUROC = 0.08, 95% CI: 0.05–0.11). LqSOFA (AUROC = 0.823, 95% CI: 0.787–0.854) demonstrated efficacy comparable to SOFA (ΔAUROC = 0.02, 95% CI: −0.02–0.06) and significantly superior to qSOFA (ΔAUROC = 0.06, 95% CI: 0.04–0.08), with a sensitivity of 0.46 (0.24–0.69)

**Data availability statement:** All analytical datasets supporting the conclusions of this article are publicly available. Specifically: All analytical datasets supporting the conclusions of this article are publicly available from the Figshare repository. They can be accessed directly via the following Digital Object Identifier (DOI): 10.6084/m9.figshare.29881934. No access restrictions apply to these datasets.

**Funding:** This research was funded by the Science and Technology Program of Zhejiang Provincial Health Commission, grant number 2021KY1016.

**Competing interests:** The authors have declared that no competing interests exist.

and specificity of 0.88 (0.80–0.93). Subgroup analyses revealed sustained high performance in both emergency department (ED) settings (AUROC = 0.82, 95% CI: 0.79–0.85) and low- and middle-income countries (LMICs) (AUROC = 0.81, 95% CI: 0.77–0.84).

## Conclusion

SOFA remains the optimal predictor of sepsis mortality risk. qSOFA demonstrates suboptimal overall predictive ability, whereas LqSOFA achieves comparable accuracy to SOFA by combining the advantages of lactate and qSOFA. Its high specificity may be valuable for rapid risk exclusion in resource-limited settings (ED/LMICs). Future studies should validate LqSOFA across diverse clinical settings and underrepresented LMIC regions and should integrate dynamic lactate clearance metrics.

## 1. Introduction

Sepsis, an organ dysfunction syndrome caused by a dysregulated host immune response triggered by infection, is one of the leading causes of death among critically ill patients worldwide, accounting for more than 11 million deaths annually [1,2]. Early and accurate prediction of the mortality risk in patients with sepsis is crucial for optimizing clinical decision-making and resource allocation. Since its proposal in 1996, the Sequential Organ Failure Assessment (SOFA) score has been the gold standard for assessing organ dysfunction and prognosis in patients with sepsis [3,4]. However, its reliance on laboratory indicators (such as blood gas analysis, bilirubin, and creatinine) limits its application in low- and middle-income countries (LMICs) with limited resources [5]. To simplify assessment, the quick Sequential Organ Failure Assessment (qSOFA), proposed in the Sepsis-3 consensus in 2016, predicts excess mortality risk through three clinical indicators: systolic blood pressure ≤ 100 mmHg, respiratory rate ≥ 22 breaths per minute, and altered mental status [1]. Although qSOFA is convenient for use in non-intensive care unit (non-ICU) settings, its sensitivity (32%–65%) and specificity (67%–94%) vary significantly across populations, with inconsistent performance especially in LMICs, where the burden of sepsis is relatively high [6–8].

In recent years, the introduction of biomarkers such as procalcitonin (PCT) and lactate has provided new perspectives for prognostic assessment. Studies have shown that PCT levels are closely related to infection severity and mortality rates [9,10]. As a key indicator of tissue hypoperfusion, an elevated lactate level can independently predict the risk of death [11]. On this basis, researchers have attempted to optimize the predictive efficacy by integrating biomarkers with qSOFA. Shetty et al. [12] confirmed in an emergency department (ED) cohort that LqSOFA≥2 (qSOFA + lactate ≥ 2 mmol/L) increased the sensitivity for predicting adverse outcomes by 17.9% compared with qSOFA alone. Yu et al. [13] and Xia et al. [14] increased the predictive sensitivity to 86.5% and 90.9%, respectively, by combining PCT with qSOFA. However, studies have significant heterogeneity in terms of

indicator combination methods, population characteristics, and outcome definitions, leading to contradictions among the results. Current guidelines give these newer indicators a low recommendation grade [1], and there is an urgent need for higher-quality evidence to support clinical practice.

This study aims to systematically evaluate the efficacy of the SOFA score, PCT, lactate level, qSOFA score, and their combined indicators (the lactate-adjusted quick Sequential Organ Failure Assessment (LqSOFA) in predicting mortality in patients with sepsis through meta-analysis. By integrating current evidence, this study provides a basis for more accurate selection of prognostic assessment tools in clinical practice and offers evidence-based medical guidance for future research directions.

## 2. Materials and methods

### 2.1. Search strategy

This study adhered to the Preferred Reporting Items for Systematic Reviews and Meta-Analyses of Diagnostic Test Accuracy Studies (PRISMA-DTA) [15] and was registered on the INPLASY platform (Registration Number: INPLASY202530075). A systematic search was conducted in the PubMed, Embase, Cochrane Library, and China National Knowledge Infrastructure (CNKI) databases (from the database inception to March 2, 2025). A combination of Medical Subject Headings (such as MeSH terms) and free terms was used in addition to Boolean logical operators (e.g., AND, OR). The search covered the following dimensions: ① study population: sepsis, suspected sepsis, severe sepsis, and septic shock; ② predictive indicators: SOFA, qSOFA, PCT, and lactate; ③ outcome indicator: mortality. No language restrictions were applied to the search, and the search was supplemented by manually screening the references of the included studies. The detailed search strategies for each database are shown in S1 Table.

### 2.2. Inclusion and exclusion criteria

The inclusion criteria were diagnostic cohort studies involving adult patients (defined as those aged ≥ 18 years) with confirmed or suspected sepsis diagnosed according to Sepsis-2.0 or Sepsis-3.0 and studies reporting the performance of SOFA, PCT, lactate, or LqSOFA for mortality prediction. Included studies needed to report the values of true positives, false positives, false negatives, and true negatives or provide the original data to calculate sensitivity, specificity, and the area under the receiver operating characteristic curve (AUROC) with a clear association between model performance (e.g., AUROC, sensitivity, specificity) and mortality outcome.

The exclusion criteria were as follows: non-diagnostic accuracy studies, methodology studies, reviews, conference abstracts, case reports, or expert consensus; studies that did not provide fourfold table data or from which sensitivity, specificity, and AUROC values could not be calculated; pediatric patients (age < 18 years); and duplicate publications (only the latest data were retained for multiple time-point reports of the same study population).

### 2.3. Literature screening and data extraction

A double-blind screening method was adopted, and researchers LJM and ZZX independently conducted the literature screening. First, obviously irrelevant studies were excluded based on titles and abstracts, followed by a secondary screening of the full texts. Eligibility was assessed according to the established inclusion/exclusion criteria. In cases of divergence, researcher GY intervened, and discussions were carried out to reach a consensus. EndNote X8 software was used for reference management. The screening results at each step, such as the total number of retrieved studies deduplicated and ultimately included, were recorded in detail, and a PRISMA flow chart was generated.

Standardized Microsoft Excel forms were used for data extraction. Extracted information included study characteristics (such as the first author, publication year, country, research design, sample size, sex, age, diagnostic criteria for sepsis), follow-up time, predictive indicators (including SOFA and qSOFA cut-off values, evaluation time points, PCT and lactate

cut-off values, and combination methods), and mortality rates. Performance indicators of the predictive model, such as the AUROC value, sensitivity, specificity, positive likelihood ratio, and negative likelihood ratio, were also collected. Researchers LJM and ZZX independently extracted the data, and inconsistencies were marked and arbitrated by researcher GY. To ensure the accuracy of data extraction, a pre-extraction test was first conducted on a small sample set, and the process was analyzed and optimized.

### 2.4. Quality assessment

The QUADAS-2 tool [16] was adopted for evaluation. The scope of evaluation covered four key areas: patient selection (i.e., consecutive inclusion/exclusion criteria), index testing (i.e., the preset threshold/operational independence), reference standards (i.e., the accuracy of the gold standard), and processing and time (i.e., the detection interval/data integrity). Two researchers independently assessed the risk level of each area and classified into three types: low, high, and unclear. Applicability assessment focused primarily on the first three items, e.g., the degree of matching with the characteristics of the sepsis population, the applicability of the adopted indicators to the research scenario of sepsis, and the consistency between the research outcomes and the sepsis-related outcomes, to judge their fit with the sepsis population, indicators, and outcomes. Disagreements between researchers LJM and GY when determining the risk level or conducting the applicability assessment were resolved through arbitration by DZZ as the third-party. The evaluation results are presented using bar charts and summary tables.

### 2.5. Statistical analysis

All analyses were performed using Stata 18.0 software, with a P value $< 0.05$ defined as statistically significant. The $2 \times 2$ contingency tables (true positives, false negatives, false positives, and true negatives) of each research report were extracted along with the values of the AUROC and the 95% CI. Sensitivity and specificity were pooled using a bivariate random-effects model [17]. Pooled positive likelihood ratio, negative likelihood ratio, and diagnostic odds ratio values were calculated. A summary receiver operating characteristic curve was fitted to evaluate the predictive efficacy of the SOFA score, PCT, lactate, qSOFA, and LqSOFA for sepsis mortality. If the predictive efficacy of multiple different indicators was evaluated in the same patient cohort (for example, the AUROCs of both SOFA score and PCT were reported simultaneously), then, based on a paired-data design, the ΔAUROC (SOFA score vs. other indicators) was pooled through the random-effects model. Heterogeneity was evaluated using the $I^2$ statistic (with a threshold of >50%) and Cochrane's Q test. When significant heterogeneity was present, a random-effects model was adopted. Forest plots were used to display the ΔAUROC and 95% CI, and the P values of the Z test were marked. Publication bias was tested using Deeks' funnel plot. Subgroup analyses were performed to explore sources of heterogeneity using the following categorical variables: (1) clinical setting (ICU vs. ED); (2) economic region (HICs/LMICs per World Bank 2024 classifications); (3) sepsis definition (Sepsis-3.0 vs. Sepsis-2.0); (4) methodology (prospective/retrospective design; sample size ≥300/<300; publication year ≥2020/<2020); (5) outcome type (28/30-day mortality vs. in-hospital/other short-term mortality); and (6) geographic region (Asian/non-Asian studies). Subgroup analysis and meta-regression were combined to explore the sources of heterogeneity, and Fagan's nomogram was applied to assist in clinical decision-making.

## 3. Results

### 3.1. Research screening process

A total of 4,504 studies were initially retrieved. After duplicates were removed, 3,868 studies remained. Through the screening of titles and abstracts, 3,724 irrelevant studies were excluded, leaving 144 for the full-text evaluation stage. Subsequently, 115 studies were excluded that did not meet the inclusion criteria (e.g., missing data, non-target population, inconsistent design). Finally, 29 studies were included. The flow chart is shown in Fig 1.

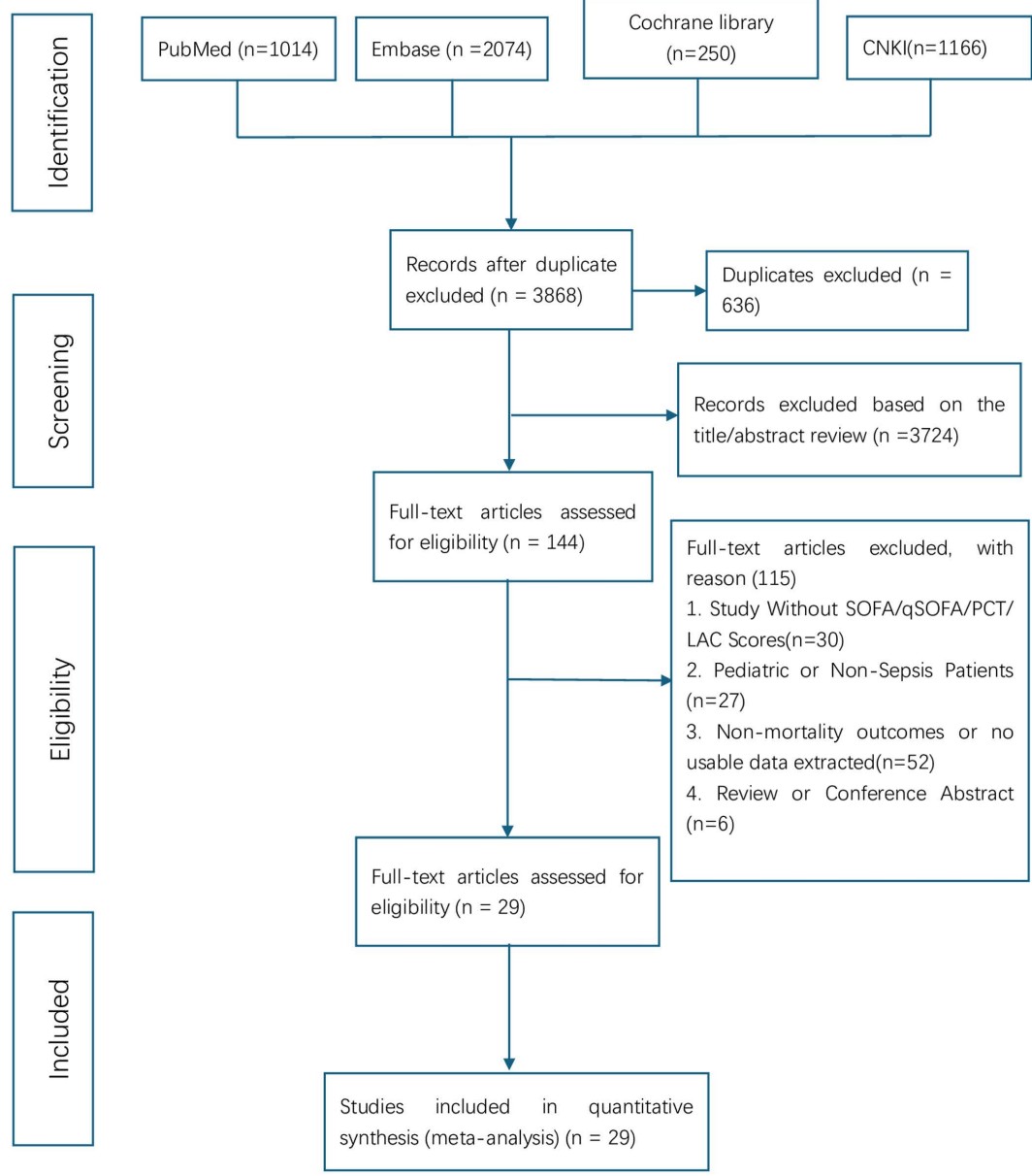

**Fig 1. The PRISMA flow chart of literature screening and selection process.**

## 3.2. Characteristics of the included studies

A total of 29 studies with publication years ranging from 2010 to 2025 were included in this meta-analysis. Ten prospective cohort studies [4,18–26] and 19 retrospective cohort studies [12–14,27–42] were included. The geographical coverage was extensive and involved multiple countries and regions, such as China (n = 12), India (n = 3), Thailand (n = 3), South Korea (n = 2), Turkey (n = 2), Spain (n = 2), Indonesia (n = 2), the United States (n = 1), Australia (n = 1), Portugal (n = 1), Vietnam (n = 1), and the Netherlands (n = 1). The total sample size was 41,469 cases with 6,293 reported deaths. The research scenarios covered the ICU (n = 14) and ED (n = 14), with one study not specifying the exact clinical setting.

Mortality rates included the 28/30-day mortality rate (n = 19), in-hospital mortality rate (n = 7), 7-day mortality rate (n = 1), ED mortality rate (n = 1), and mortality rate within 72 hours (n = 1). The diagnostic criteria for sepsis in the included studies were Sepsis-2.0 (n = 6 studies) and Sepsis-3.0 (n = 23 studies). With regard to the economic context, 22 studies (75.9%) originated from LMICs, while 7 studies (24.1%) were from HICs. Detection indicators included lactate, PCT, SOFA, qSOFA, and LqSOFA. The distribution of studies evaluating each indicator and their pairwise overlaps (e.g., cohorts enabling direct AUROC comparisons) are quantified in S1 Fig. Different cutoff values were set for each indicator, and the evaluation time was concentrated on key periods, such as within 24 hours of admission, at ICU admission, and during ED assessment. To distinguish the content of the Derivation Cohort External and Validation Cohort reported in two independent studies, namely, Wright, S. W. (2022) [25] and Li, F. (2023) [33], the derivation cohort external of Wright, S. W. (2022) was defined as Wright, S. W. 2022a, and its validation cohort was defined as Wright, S. W. 2022b, and the derivation cohort external of Li, F. (2023) was defined as Li, F. 2023a, and its validation cohort was defined as Li, F. 2023b. The detailed characteristics of the included studies are shown in Tables 1 and 2.

### 3.3. Quality assessment

A total of 29 studies were evaluated using QUADAS-2 [16]. In terms of patient selection, 27 studies were judged as "yes", indicating a low risk of bias. Two studies were judged as "uncertain" due to the lack of description of the exclusion criteria and issues with the enrollment method. In the field of index detection, all studies were judged as "yes", indicating a low risk of bias in this area. Similarly, in the field of reference standards, all studies were judged as "yes", indicating a low risk of bias. In terms of process and time, 6 studies were judged as "uncertain" due to the lack of description of the detection and evaluation time, and 4 studies were judged as "uncertain" due to the omission of the patient evaluation process. Overall, the reference standard field fully met the criteria. Most patient selection and index detection methods are standardized, and the process and time are generally controllable. The overall bias of all studies was mostly low, and the applicability was high. The included studies were high quality and had a good correlation with clinical practice. For details, refer to S2–4 Figs.

### 3.4. Diagnostic efficacy indicators

#### 3.4.1. Diagnostic efficacy of PCT.
PCT demonstrated moderate diagnostic value for sepsis mortality in 12 studies (n = 6776) with a sensitivity of 0.76 (95% CI: 0.65–0.84) and specificity of 0.65 (95% CI: 0.53–0.75), with an AUROC of 0.764 (0.725–0.800) (Table 3, Fig 2). Fagan's nomogram indicated that the positive posterior probability increased to 35% (vs. 20% pretest), while the negative probability decreased to 8% (Fig 3). Significant heterogeneity was observed (both p < 0.01), with sensitivity heterogeneity associated with the sepsis definition (p < 0.01) and specificity heterogeneity associated with publication year (p < 0.05). Subgroup analyses aligned with the overall results. No publication bias was detected (p = 0.11). For details, see S5–7 Figs and S2 Table.

#### 3.4.2. Diagnostic efficacy of Lactate.
Lactate demonstrated moderate diagnostic value for sepsis mortality in 14 studies (n = 14,485) with a sensitivity of 0.68 (95% CI: 0.58–0.76) and specificity of 0.69 (95% CI: 0.62–0.75), and AUROC of 0.740 (0.700–0.777) (Table 3, Fig 2). Fagan's nomogram indicated a positive posterior probability of 36% (vs. 20% pretest) and a negative posterior probability of 10% (Fig 3). Significant heterogeneity was observed (both p < 0.01), with specificity heterogeneity linked to study design and sepsis definition (both p < 0.01). Subgroup analyses revealed that after 3 prospective studies were excluded, sensitivity increased to 0.82 (0.65–0.92) in small-sample studies (<300 patients). Other subgroup results were consistent with the overall findings. No publication bias was detected (p = 0.10). For details, see S5–7 Figs and S3 Table.

#### 3.4.3. Diagnostic efficacy of qSOFA.
The qSOFA score showed low diagnostic value for sepsis mortality in 14 studies (n = 30,137) with a sensitivity of 0.52 (95% CI: 0.33–0.71), specificity of 0.77 (95% CI: 0.64–0.86), and AUROC of

**Table 1. Characteristics of include studies.**

| ID | Author | Year | Country | Study design | Set-ting | Type of mortality | Sample (N) | Died (%) | Age | Detection index |
|---|---|---|---|---|---|---|---|---|---|---|
| 1 | Suárez-Santamaría, M. | 2010 | Spain | Prospective | ICU | 28 – day mortality | 253 | 55 | 65(range 19–94) | Lactate/PCT/SOFA |
| 2 | Wang, J. | 2016 | China | Retrospective | ICU | In-hospital mortality | 864 | 132 | 63.56 ± 15.80 | SOFA/PCT |
| 3 | Shetty, A. | 2017 | Austra-lia, the Netherlands | Retrospective | ED | Mortality and/or ICU stay ≥72 h) | 12555 | 572 | The median age of patients across centers ranged from 47 to 72.4 years | SOFA/qSOFA/LqSOFA |
| 4 | Zhao, R. | 2017 | China | Prospective | ICU | 28 – day mortality | 104 | 26 | SIRS group: 58.8 ± 11.3; Sepsis group: 57.6 ± 12.9 | PCT/SOFA |
| 5 | Liu, Z. | 2019 | USA | Retrospective | ICU | 30-day mortality | 1865 | 809 | 68 (IQR: 56–78.25) | Lactate/qSOFA/SOFA/LqSOFA |
| 6 | Yu, H. | 2019 | China | Retrospective | ED | 30-day mortality | 1318 | 178 | 64 (IQR: 47–75) | qSOFA/PqSOFA |
| 7 | Zhang, Y. | 2019 | China | Retrospective | ICU | 28-day mortality | 150 | 98 | Control: 66.5 (25–89); Sepsis: 70 (24–91); Septic shock: 74.5 (24–89) | PCT/SOFA/qSOFA |
| 8 | Sinto, R. | 2020 | Indonesia | Prospective | ED | 28-day mortality | 1213 | 421 | 51 (IQR: 38–60) | SOFA/Lactate/qSOFA/LqSOFA |
| 9 | Xia, Y. | 2020 | China | Retrospective | ED | 28-day mortality | 821 | 173 | Survival: 56.83 ± 17.79; Deceased: 59.92 ± 17.79 | Lactate/PCT/qSOFA/PqSOFA/SOFA |
| 10 | Zhou, H. | 2020 | China | Retrospective | ED | 28-day mortality | 336 | 89 | 76 (IQR: 61–84) | Lactate/SOFA/qSOFA/LqSOFA/LSOFA |
| 11 | Liu, S. | 2020 | China | Retrospective | ED | In-hospital mortality | 821 | 173 | 58.3 ± 17.09 | LqSOFA/qSOFA |
| 12 | Daga, M. K. | 2021 | India | Prospective | ICU | 7-day mortality | 150 | 73 | 48.46 ± 15.28 | SOFA/qSOFA/Lactate/LqSOFA |
| 13 | Hao, C. | 2021 | China | Retrospective | ICU | 28-day mortality | 303 | 179 | 64.36 ± 15.62 | Lactate/PCT/SOFA |
| 14 | Kilinc Toker, A. | 2021 | Turkey | Retrospective | ED | ED mortality | 499 | 115 | 72.5 ± 13.7 | SOFA/qSOFA/LqSOFA |
| 15 | Suttapanit, K. | 2021 | Thailand | Prospective | ED | 28-day mortality | 1139 | 118 | Survival: 70 (69–71); Deceased: 70 (67–73) | LqSOFA/qSOFA/SOFA |
| 16 | Guarino, M. | 2022 | Italy | Retrospective | ED | In-hospital mortality | 556 | 218 | 79.9 ± 11.9 | Lactate/qSOFA |
| 17 | Jaiswal, P. | 2022 | India | Retrospective | ICU | In-hospital mortality | 280 | 121 | 59.38 ± 15.88 | Lactate/SOFA |
| 18 | Sen, P. | 2021 | Turkey | Retrospective | ICU | 28-day mortality | 165 | 119 | 60.9 ± 17.9 | PCT/SOFA |
| 19 | Silva, C. M. | 2022 | Portugal | Prospective | ICU | In-hospital mortality | 1640 | NR | NR | Lactate/SOFA |
| 20 | Wang, L. | 2022 | China | Prospective | ED | 28-day mortality | 175 | 41 | 66 (IQR: 53–77) | Lactate/PCT/qSOFA/LqSOFA |
| 21a | Wright, S. W. | 2022 | Thailand, Vietnam, Indonesia | Prospective | NR | 28-day mortality | 4980 | 816 | 57 (IQR: 41–71) | qSOFA/SOFA/Lactate/LqSOFA |
| 21b | | | | | | | 792 | 102 | 51 (IQR: 33–65) | qSOFA/SOFA/Lactate/LqSOFA |

*(Continued)*

**Table 1.** (Continued)

| ID | Author | Year | Country | Study design | Setting | Type of mortality | Sample (N) | Died (%) | Age | Detection index |
|---|---|---|---|---|---|---|---|---|---|---|
| 22 | Chi, H. | 2023 | China | Retrospective | ICU | 28-day mortality | 135 | 62 | 71.47 ± 15.36 | SOFA/Lactate |
| 23 | Julián-Jiménez, A. | 2023 | Spain | Prospective | ED | 30-day mortality | 4439 | 459 | 67 ± 18 | Lactate/qSOFA/LqSOFA |
| 24a | Li, F. | 2023 | China | Retrospective | ICU | 28-day mortality | 340 | 102 | 56.00 (IQR: 44.25–68.00) | Lactate/PCT/SOFA/LSOFA |
| 24b | | | | | | | 75 | 37 | 61.00 (IQR: 44.00–71.00) | PSOFA/Lactate/PCT/SOFA |
| 25 | Noparatkailas, N. | 2023 | Thailand | Retrospective | ED | 28-day mortality | 448 | 99 | 71 (IQR: 59–87) | Lactate/qSOFA/LqSOFA |
| 26 | Shinde, V. V. | 2023 | India | Prospective | ICU | In-hospital mortality | 80 | 30 | 58 ± 12 | PCT/SOFA |
| 27 | Li, L. | 2024 | China | Retrospective | ICU | 28-day mortality | 200 | 67 | NR | SOFA/PCT |
| 28 | Yoo, K. H. | 2024 | South Korea | Retrospective | ED | 28-day mortality | 3499 | 792 | 70 (IQR: 61–78) | Lactate/SOFA/qSOFA/PCT |
| 29 | Saqer M Althunayyan | 2025 | Saudi Arabia | Retrospective | ED | Mortality within 72h | 1274 | 17 | 68.80 ± 17.9 | qSOFA/LqSofa |

Abbreviations: ICU, Intensive Care Unit; PCT, Procalcitonin; SOFA, Sequential Organ Failure Assessment; ED, Emergency Department; qSOFA, Quick Sequential Organ Failure Assessment; LqSOFA, Lactate-adjusted Quick Sequential Organ Failure Assessment; PqSOFA, Procalcitonin-adjusted Quick Sequential Organ Failure Assessment; LSOFA, Lactate-adjusted Sequential Organ Failure Assessment; NR, Not Recorded;

0.721 (0.680–0.759) (Table 3, Fig 2). Fagan's analysis indicated a positive posterior probability of 36% (vs. 20% pretest) and a negative probability of 13% (Fig 3). Significant heterogeneity was observed (both p < 0.01; $I^2 > 99\%$), although meta-regression revealed no significant moderators. Subgroup analyses showed notable variations in sensitivity across study designs and outcome subgroups. No publication bias was detected (p = 0.71). For details, see S5–7 Figs and S4 Table.

**3.4.4. Diagnostic efficacy of LqSOFA.** LqSOFA demonstrated superior diagnostic value for sepsis mortality in 9 studies (n = 22,078), with a sensitivity of 0.46 (95% CI: 0.24–0.69), specificity of 0.88 (95% CI: 0.80–0.93), and AUROC of 0.823 (0.787–0.854) (Table 3, Fig 2). Fagan's analysis indicated a positive posterior probability of 49% (vs. 20% pretest) and a negative probability of 13% (Fig 3). Significant heterogeneity was observed (both p < 0.01; $I^2 > 97\%$), with no significant moderators identified by meta-regression. No publication bias was detected (p = 0.71). For details, see S5–7 Figures.

**3.4.5. Subgroup analysis of LqSOFA diagnostic performance.** For sepsis mortality prediction, LqSOFA showed consistently robust performance. Subgroup analyses by both sepsis definition and clinical setting were precluded due to homogeneity across all included studies: all studies (N = 9) exclusively used Sepsis-3 criteria and were conducted in ED settings. Similarly, in LMICs (7 studies), the pooled AUC reached 0.81 (95% CI: 0.77–0.84) with a sensitivity of 0.41 (95% CI: 0.17–0.70) and specificity of 0.88 (0.77–0.94). Subsequent analyses revealed improved 28/30-day mortality prediction (6 studies: AUC 0.85 [95% CI: 0.82–0.88], sensitivity 0.60 [0.50–0.69], specificity 0.86 [0.83–0.89]). Prospective studies (n = 4) showed higher sensitivity (0.64 [0.56–0.71] vs. 0.31 [0.09–0.68]) but comparable specificity (0.86 [0.81–0.90] vs. 0.90 [0.74–0.96]) compared to retrospective designs (n = 5). ICU, non-Asian, small-scale (<300), and pre-2020 subgroups had insufficient data for pooled estimates. Detailed subgroup analysis findings are detailed in Table 4.

**3.4.6. Diagnostic efficacy of SOFA.** The SOFA score demonstrated high diagnostic value for sepsis mortality in 18 studies (n = 23,802), with a sensitivity of 0.77 (95% CI: 0.71–0.82), specificity of 0.73 (95% CI: 0.67–0.79), and AUROC of 0.819 (0.783–0.850) (Table 3, Fig 2). Fagan's analysis indicated a positive posterior probability of 42% (vs. 20% pretest) and a negative probability of 7% (Fig 3). Significant heterogeneity was observed (both p < 0.01; $I^2 > 96\%$). Meta-regression identified multiple modifiers for sensitivity (publication year, economic level, setting, sample fraction, sepsis definition) and specificity

**Table 2. Diagnostic and Detection Characteristics of Included Studies.**

| ID | Author | Year | Sepsis diagnosis | Detection index & cut-off value | Assessment/Detection Time |
|---|---|---|---|---|---|
| 1 | Suárez-Santamaría, M. | 2010 | Sepsis was defined as clinical infection evidence plus ≥2 SIRS criteria. (Sepsis-2) | Lactate: NR; PCT: NR; SOFA: NR | At admission |
| 2 | Wang, J. | 2016 | Diagnostic criteria refer to the "Diagnostic Criteria for Nosocomial Infection" issued by the Family Planning Commission in 2001 (Trial). (Sepsis-2) | SOFA: 6.37; PCT: 3.38 µg/L | Within 24 hours of ICU admission |
| 3 | Shetty, A. | 2017 | Adult patients with suspected or proven sepsis presenting to EDs, identified by SIRS screening. (Sepsis-3) | SOFA: ≥2; qSOFA: ≥2; LqSOFA: ≥2 | Worst values during ED stay |
| 4 | Zhao, R. | 2017 | Adult patients with sepsis or systemic inflammatory response syndrome (SIRS). (Sepsis-2) | PCT: 7.68 µg/L; SOFA: 12.5 | Within 24 hours of ICU admission |
| 5 | Liu, Z. | 2019 | Adult Patients who were diagnosed with 'sepsis', 'severe sepsis' and 'septic shock' on discharge (Sepsis-3) | Lactate: 3.225 mmol/L; qSOFA: ≥2; SOFA: NR; LqSOFA: NR | Within 24 hours of ICU admission |
| 6 | Yu, H. | 2019 | Adult patients presenting to ED/hospital with systemic infection symptoms. (Sepsis-3) | qSOFA: ≥2; PqSOFA: ≥2 | Within 24 hours of admission |
| 7 | Zhang, Y. | 2019 | Adult patients diagnosed with sepsis or septic shock based on sepsis-3 criteria. (Sepsis-3) | PCT: 4.7; SOFA: 4; qSOFA: NR | At ICU admission |
| 8 | Sinto, R. | 2020 | Adult patients with suspected bacterial infection (on antibiotics and with cultures). (Sepsis-3) | SOFA: ≥2; Lactate: >2 mmol/L; qSOFA: ≥2; LqSOFA: qSOFA≥2 and Lactate >2 mmol/L | Worst values 12 hours before enrolment |
| 9 | Xia, Y. | 2020 | Patients >14 years old, treated at ED, and meeting Sepsis 2.0 criteria. (Sepsis-2) | Lactate: 2.35; PCT: 0.51; qSOFA: 2; PqSOFA: 2; SOFA: NR | Within 24 hours of admission |
| 10 | Zhou, H. | 2020 | new chest x-ray infiltrates, ≥2 symptoms (cough, fever, dyspnea, etc.), and CAP patients with SOFA score increase ≥2 according to sepsis 3.0. | Lactate: 2; SOFA: 4; qSOFA: 2; LqSOFA: 0.29 (probability threshold); LSOFA: 0.23 (probability threshold) | At admission |
| 11 | Liu, S. | 2020 | Per sepsis-3, patients enrolled had infection-induced SOFA score increase ≥2. (Sepsis-3) | LqSOFA: NR; qSOFA: NR | At admission |
| 12 | Daga, M. K. | 2021 | Patients suspected of sepsis based on SEPSIS 3 guidelines (2016). (Sepsis-3) | SOFA: 8.5; Lactate: ≥2 mmol/L; qSOFA: NR; LqSOFA: NR | Within 24 hours of admission |
| 13 | Hao, C. | 2021 | Patients with septic shock hospitalized in the department of critical care medicine. (Sepsis-3) | Lactate: 3.55; PCT: 14.2; SOFA: 7.5; | Immediately after ICU admission |
| 14 | Kilinc Toker, A. | 2021 | For patients with sepsis, the diagnostic basis is not described in detail. (Sepsis-3) | SOFA:>11; qSOFA:>1; LqSOFA:>3 | During emergency assessment |
| 15 | Suttapanit, K. | 2021 | Patients 18 years and older who visited ED with suspected sepsis. (Sepsis-3) | LqSOFA: ≥3; qSOFA: ≥2; SOFA: ≥2 | At admission |
| 16 | Guarino, M. | 2022 | patients identified by 'sepsis' and 'septic shock' in discharge letter. (Sepsis-3) | Lactate: ≥1.85 mmol/L; qSOFA: ≥2 | At first ED assessment |
| 17 | Jaiswal, P. | 2022 | patients diagnosed with SIRS, Sepsis, Severe Sepsis, and septic shock. (Sepsis-2) | Lactate: >3; SOFA: 9 | At admission (Lactate); Within 24 hours of ICU admission (SOFA) |
| 18 | Sen, P. | 2021 | Sepsis identified per Sepsis-3 definition. (Sepsis-3) | PCT: 0.8ng/mL; SOFA: 7 | Within 24 hours of ICU admission |
| 19 | Silva, C. M. | 2022 | Sepsis identified per Sepsis-2 definition. (Sepsis-2) | Lactate: NR; SOFA: NR | Within 12 hours of admission (Lactate); At admission (SOFA) |
| 20 | Wang, L. | 2022 | Sepsis identified per Sepsis-3 definition. (Sepsis-3) | Lactate: 1.855 mmol/L; PCT: 9.24 ng/mL; qSOFA: 1.5; LqSOFA: NR; PqSOFA: NR | Within 24 hours of admission (Lactate); At ED admission (qSOFA) |
| 21a/b | Wright, S. W. | 2022 | Patients with ≥3 systemic manifestations (2012 Surviving Sepsis Campaign). (Sepsis-3) | qSOFA: NR; SOFA: NR; Lactate: NR; LqSOFA: NR | Calculated at enrolment (qSOFA/SOFA); Point-of-care measurement at enrolment (Lactate) |

*(Continued)*

**Table 2.** (Continued)

| ID | Author | Year | Sepsis diagnosis | Detection index & cut-off value | Assessment/Detection Time |
|---|---|---|---|---|---|
| 22 | Chi, H. | 2023 | Sepsis identified per Sepsis-3 definition. (Sepsis-3) | SOFA: 8.5; Lactate: 2.55 | Immediately after transfer to ICU |
| 23 | Julián-Jiménez, A. | 2023 | Each patient diagnosed with suspected infection based on epidemiology. (Sepsis-3) | Lactate: ≥ 2 mmol/L; qSOFA: 2; LqSOFA: qSOFA≥2 + Lactate ≥2 | At ED presentation (Lactate); At admission (qSOFA) |
| 24a /b | Li, F. | 2023 | Sepsis identified per Sepsis-3 definition. (Sepsis-3) | Lactate: 2.4; PCT: 8.03; SOFA: ≥ 6; Lac+SOFA: 2.4 | Within 24 hours of ICU admission |
| 25 | Noparatkailas, N. | 2023 | Sepsis defined as suspected/confirmed infection with SIRS/qSOFA ≥2. (Sepsis-3) | Lactate: ≥ 2 mmol/L; qSOFA: ≥ 2; LqSOFA: ≥ 2 mmol/L + Lactate ≥ 2 | Initial serum lactate at ED (Lactate); At admission (qSOFA) |
| 26 | Shinde, V. V. | 2023 | Sepsis identified per Sepsis-3 definition. (Sepsis-3) | PCT: 4.15; SOFA: 8 | At admission |
| 27 | Li, L. | 2024 | Sepsis identified per Sepsis-3 definition. (Sepsis-3) | SOFA: NR; PCT: 4.05 | At admission |
| 28 | Yoo, K. H. | 2024 | Patients with suspected/inconfirmed infection and refractory hypotension/hyperlactatemia. (Sepsis-3) | Lactate: 1.517; SOFA: 7.5; PCT: 4.517; qSOFA: NR | Time zero of sepsis (Lactate); At admission (qSOFA) |
| 29 | Saqer M Althunayyan | 2025 | Suspected sepsis: blood culture during ED visit and IV antibiotics administered. (Sepsis-3) | qSOFA: 2; LqSofa: 2 | Calculated using initial ED triage data |

Abbreviations: SIRS, Systemic Inflammatory Response Syndrome; NR, Not Recorded; SOFA, Sequential Organ Failure Assessment; ICU, Intensive Care Unit; ED, Emergency Department; qSOFA, Quick Sequential Organ Failure Assessment; LqSOFA, Lactate-adjusted Quick Sequential Organ Failure Assessment; PqSOFA, Procalcitonin-adjusted Quick Sequential Organ Failure Assessment; LSOFA, Lactate-adjusted Sequential Organ Failure Assessment.

**Table 3. Pooled Performance of PCT, Lactate, qSOFA, LqSOFA, and SOFA in Predicting Sepsis Patient Mortality.**

| | No of studies | SROC | Sensitivity | Specificity | PLR | NLR | DOR |
|---|---|---|---|---|---|---|---|
| **PCT** | 12 | 0.764 [0.725, 0.800] | 0.76 [0.65, 0.84] | 0.65 [0.53, 0.75] | 2.2 [1.6, 3.0] | 0.37 [0.25, 0.56] | 6 [3,11] |
| **Lactate** | 14 | 0.740 [0.700, 0.777] | 0.68 [0.58, 0.76] | 0.69 [0.62, 0.75] | 2.2 [1.9, 2.5] | 0.46 [0.37, 0.58] | 5 [4,6] |
| **qSOFA** | 14 | 0.721 [0.680, 0.759] | 0.52 [0.33, 0.71] | 0.77 [0.64, 0.86] | 2.2 [1.8, 2.8] | 0.62 [0.46, 0.85] | 4 [2,5] |
| **LqSOFA** | 9 | 0.823 [0.787, 0.854] | 0.46 [0.24, 0.69] | 0.88 [0.80, 0.93] | 3.8 [2.7, 5.3] | 0.62 [0.42, 0.91] | 6 [3,11] |
| **SOFA** | 18 | 0.819 [0.783, 0.850] | 0.77 [0.71, 0.82] | 0.73 [0.67, 0.79] | 2.9 [2.3, 3.5] | 0.31 [0.25, 0.39] | 9 [7,13] |

Abbreviations: qSOFA, Quick Sequential Organ Failure Assessment; LqSOFA, Lactate-adjusted Quick Sequential Organ Failure Assessment; SOFA, Sequential Organ Failure Assessment; PLR, Positive Likelihood Ratio; NLR, Negative Likelihood Ratio; DOR, Diagnostic Odds Ratio; SROC, Summary Receiver Operating Characteristic.

(publication year, economic level, sample fraction, sepsis definition) heterogeneity (all p < 0.05). Subgroup analyses revealed stable results across subgroups. Publication bias was detected (p = 0.03). For details, see S5–7 Figs and S5 Table.

**3.4.7. Comparison of the differences in AUROC among SOFA, PCT, LAC, qSOFA, and LqSOFA.** Based on the meta-analysis of the AUROC differences for paired data, the AUROC of SOFA was 0.10 higher than procalcitonin (n = 12, 95% CI: 0.04–0.16; $I^2$ = 84.7%), 0.07 higher than lactate (n = 15, 95% CI: 0.03–0.11; $I^2$ = 88.7%), and 0.08 higher than qSOFA (n = 11, 95% CI: 0.05–0.11; $I^2$ = 83.5%). In contrast, LqSOFA showed an AUROC increase of 0.06 versus qSOFA (n = 14, 95% CI: 0.04–0.08; $I^2$ = 59.8%). However, there was no significant difference between SOFA and LqSOFA scores, with a difference of 0.02 (n = 8, 95% CI: −0.02–0.06; $I^2$ = 89.2%) (Fig 4).

## 4. Discussion

Our study systematically assessed the predictive performance of PCT, lactate, qSOFA, LqSOFA, and SOFA for mortality risk in sepsis patients through a meta-analysis. The results demonstrated that the SOFA score (AUROC 0.819)

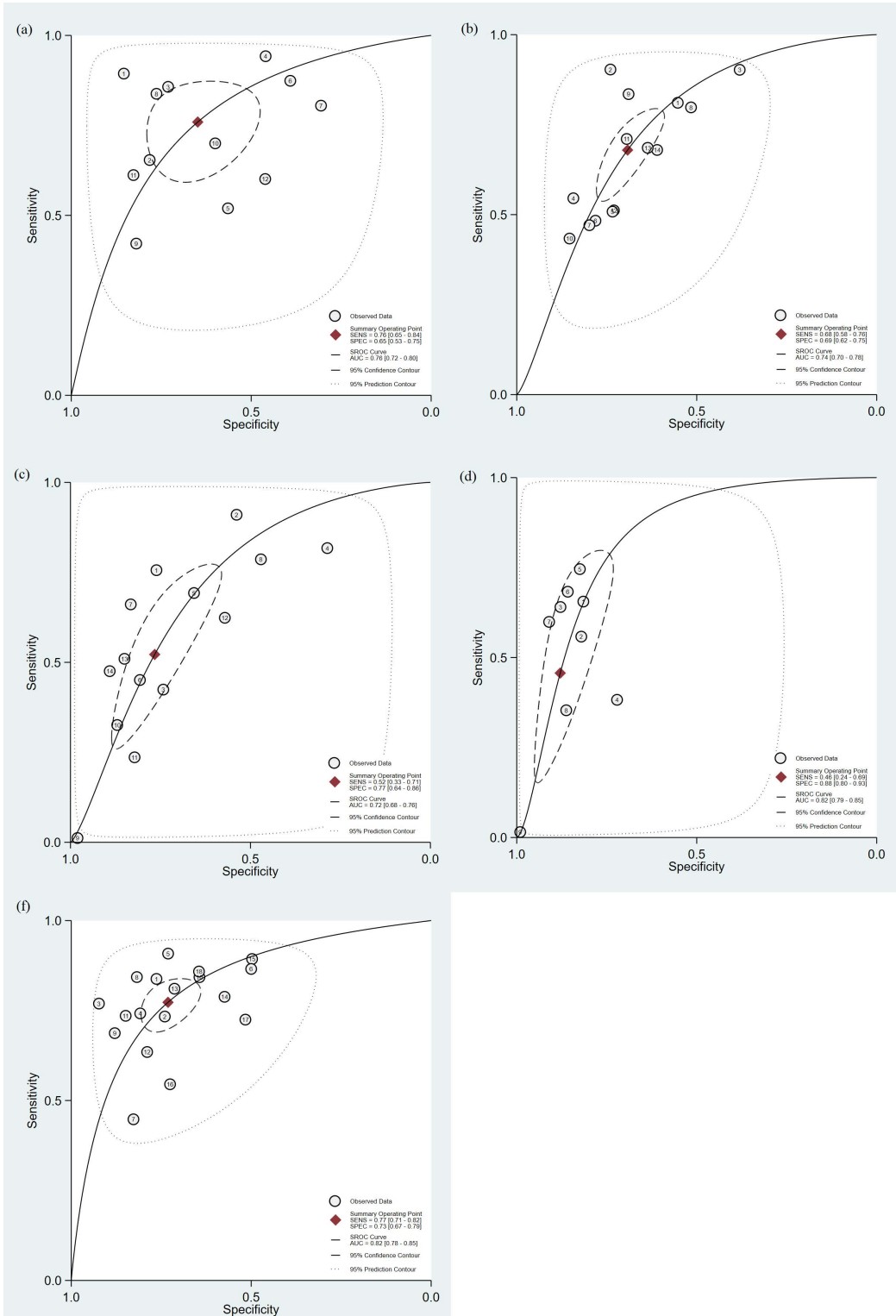

**Fig 2. HSROC curve for predicting mortality in patients with sepsis.** (a) PCT; (b) Lactate; (c)) qSOFA; (d) Lqsofa; (f) Sofa.

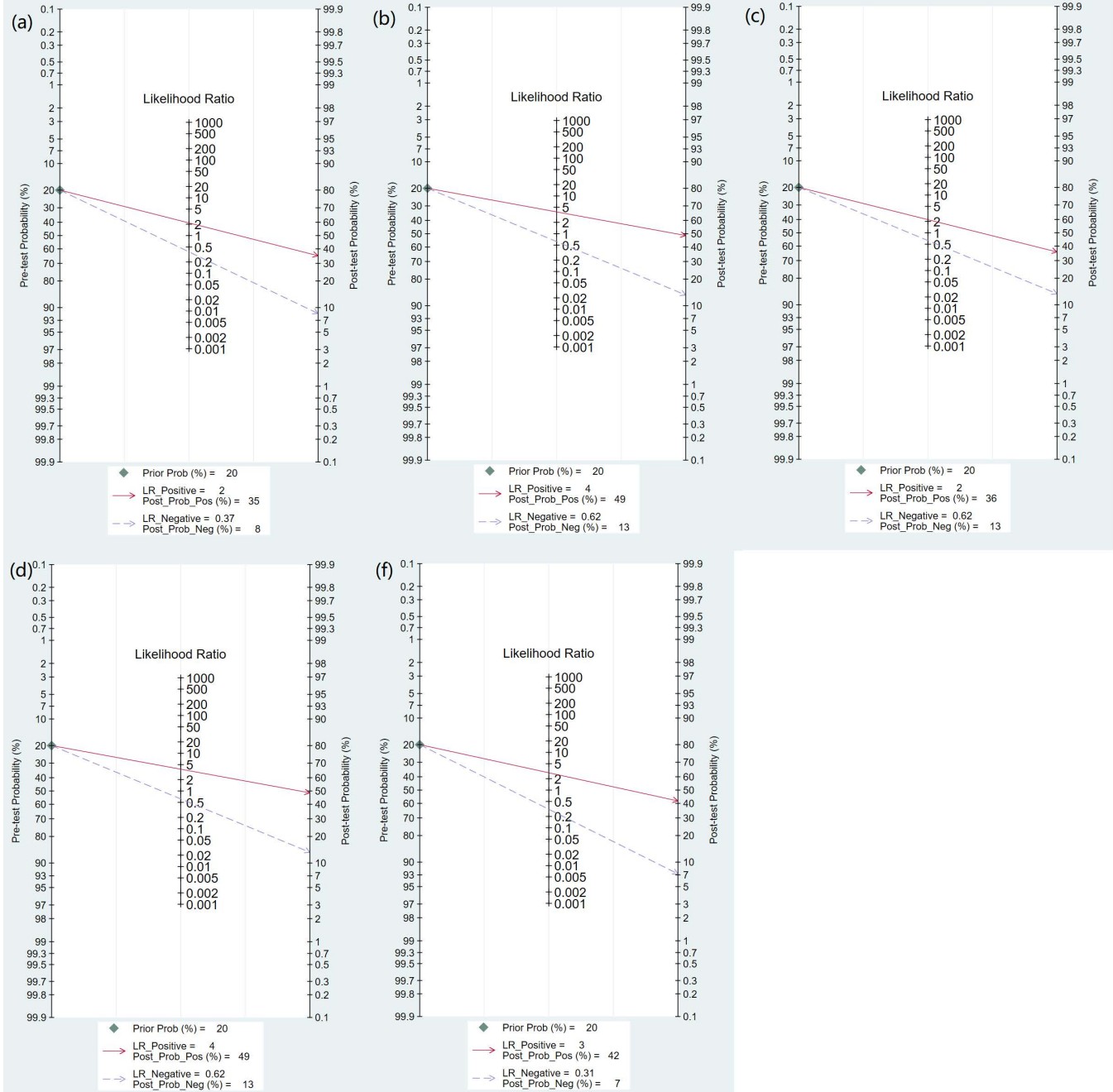

**Fig 3. Fagan nomogram of pretest probability and negative posttest probability.** (a) PCT; (b) Lactate; (c))qSOFA; (d)Lqsofa; (f)Sofa.

significantly outperformed PCT (ΔAUROC 0.10), lactate (ΔAUROC 0.07), and qSOFA (ΔAUROC 0.08) in terms of predictive performance. Notably, LqSOFA exhibited comparable predictive power to SOFA (ΔAUROC = 0.02; AUROC = 0.823) and a clinically relevant improvement over qSOFA (ΔAUROC 0.06). Its high specificity (0.88) further demonstrated unique clinical value for rapidly ruling out patients with low mortality risk. Subgroup analyses further confirmed that LqSOFA

**Table 4. Subgroup Analyses of Pooled Diagnostic Performance of LqSOFA in Predicting Sepsis Patient Mortality.**

| Subgroup Variables | Group Definition | No of studies | SROC | Sensitivity | Specificity | PLR | NLR | DOR |
|---|---|---|---|---|---|---|---|---|
| Setting | ICU | 0 | — | — | — | — | — | — |
| | ED | 9 | 0.82[0.79, 0.85] | 0.46 [0.24, 0.69] | 0.88 [0.80, 0.93] | 3.8 [2.7, 5.3] | 0.62 [0.42, 0.91] | 6 [3,11] |
| Income Group | HICs | 2 | — | — | — | — | — | — |
| | LMICs | 7 | 0.81[0.77, 0.84] | 0.41 [0.17, 0.70] | 0.88 [0.77, 0.94] | 3.4 [2.3, 5.2] | 0.67 [0.44, 1.03] | 5 [2,10] |
| Sepsis criteria | Sepsis-3 | 9 | 0.82[0.79, 0.85] | 0.46 [0.24, 0.69] | 0.88 [0.80, 0.93] | 3.8 [2.7, 5.3] | 0.62 [0.42, 0.91] | 6 [3,11] |
| | Sepsis-2 | 0 | — | — | — | — | — | — |
| Publish year | ≥2020 | 8 | 0.82[0.79, 0.85] | 0.43 [0.20, 0.69] | 0.89 [0.79, 0.94] | 3.8 [2.6, 5.6] | 0.64 [0.43, 0.97] | 6 [3,12] |
| | <2020 | 1 | — | — | — | — | — | — |
| Region | Asia | 7 | 0.81[0.77, 0.84] | 0.41 [0.17, 0.70] | 0.88 [0.77, 0.94] | 3.4 [2.3, 5.2] | 0.67 [0.44, 1.03] | 5 [2,10] |
| | Non-Asia | 2 | — | — | — | — | — | — |
| Study design | Prospective | 4 | 0.82[0.79, 0.86] | 0.64 [0.56, 0.71] | 0.86 [0.81, 0.90] | 4.5 [3.4, 6.0] | 0.42 [0.34, 0.52] | 11 [7,16] |
| | Retrospective | 5 | 0.78[0.74, 0.81] | 0.31 [0.09, 0.68] | 0.90 [0.74, 0.96] | 3.0 [1.7, 5.1] | 0.77 [0.53, 1.13] | 4 [2,9] |
| Outcome | 28/30-day mortality | 6 | 0.85[0.82, 0.88] | 0.60 [0.50, 0.69] | 0.86 [0.83, 0.89] | 4.4 [3.4, 5.6] | 0.47 [0.36, 0.60] | 9 [6,15] |
| | Other mortality | 3 | — | — | — | — | — | — |
| Sample size | ≥300 | 8 | 0.82[0.78, 0.85] | 0.43 [0.20, 0.69] | 0.88 [0.79, 0.94] | 3.7 [2.5, 5.3] | 0.65 [0.43, 0.96] | 6 [3,11] |
| | <300 | 1 | — | — | — | — | — | — |

Abbreviations: LqSOFA, Lactate-adjusted Quick Sequential Organ Failure Assessment; SROC, Summary Receiver Operating Characteristic; PLR, Positive Likelihood Ratio; NLR, Negative Likelihood Ratio; DOR, Diagnostic Odds Ratio; ICU, Intensive Care Unit; ED, Emergency Department; HICs, High-Income Countries; LMICs, Low- and Middle-Income Countries.

demonstrated robust predictive ability across ED settings (AUROC 0.82) and LMICs (AUROC 0.81) and in predicting 28/30-day mortality (AUROC 0.85), providing an evidence-based rationale for streamlining clinical assessment workflows.

This meta-analysis confirmed that the SOFA score remains the gold standard for predicting sepsis mortality, supported by its comprehensive multisystem assessment and alignment with prior evidence. Qiu et al.'s systematic review further validated SOFA's optimal performance (sensitivity 0.89/specificity 0.69) for in-hospital mortality, particularly in 28/30-day screening within resource-limited settings [43]. Notably, qSOFA demonstrated significant predictive limitations in our cohort, with a sensitivity of only 0.52 (95% CI: 0.33–0.71) and a specificity of 0.77 (95% CI: 0.64–0.86), indicating insufficient standalone clinical utility. This finding aligns closely with the Sepsis-3 guidelines, which caution that qSOFA, when used as an isolated screening tool, has sensitivity deficiencies and should be combined with other biomarkers to optimize predictive performance [1]. Further analysis revealed that LqSOFA achieves comparable predictive performance to SOFA (AUROC=0.823, ΔAUROC=0.02) by synergistically integrating the advantages of lactate as a tissue perfusion marker with the value of qSOFA as an organ dysfunction screening tool. These findings align with Moncada-Gutierrez et al.'s meta-analysis (AUROC = 0.807, n = 23,551) [44]. Critically, in our study, the high specificity of LqSOFA (0.88, 95% CI: 0.80–0.93) provided an exceptionally low-risk patient exclusion capability, with negative predictive values consistently exceeding 90%.

Our subgroup analysis revealed the clinical utility of LqSOFA across diverse medical contexts. Among the 9 studies conducted in ED settings, LqSOFA demonstrated robust predictive performance (AUROC 0.82, 95% CI: 0.79–0.85; specificity 0.88, 95% CI: 0.80–0.93). Its high specificity significantly optimized triage decisions, yielding negative predictive

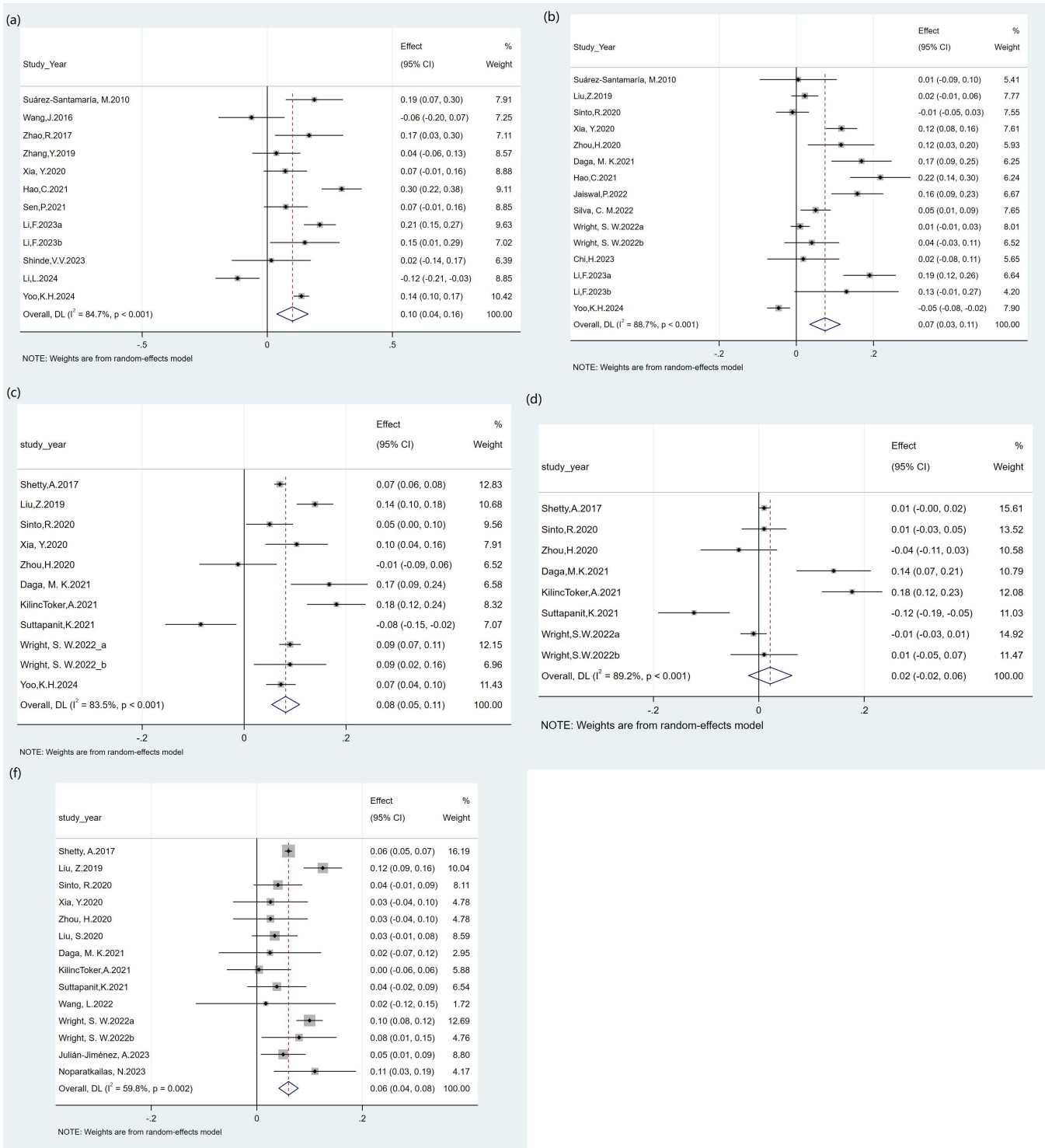

**Fig 4. Forest Plot of Pooled AUROC Differences.** (a) Sofa vs PCT; (b) Sofa vs Lactate; (c)) Sofa vs qSOFA; (d) Sofa vs Lqsofa; (f) Lqsofa vs qsofa.

values >90%. This conclusion is supported by multiple prospective studies: Kilinc Toker et al. reported a 47.8% reduction in ICU assessment demand through LqSOFA implementation [32], whereas Sinto et al. validated its ability to decrease ICU misdirected transfer rates by 39% in resource-limited settings, demonstrating performance parity with SOFA [21]. By enabling precise exclusion of low-risk patients, LqSOFA provides an efficient solution for emergency department triage. Across 7 studies in LMICs, LqSOFA maintained robust predictive ability (AUROC 0.81). Its core advantage lies in minimizing laboratory dependency: qSOFA components require only a sphygmomanometer and timer for bedside assessment, whereas lactate measurements can be rapidly obtained via portable devices or point-of-care arterial blood gas analyzers with a median turnaround time of just 15 minutes. This represents a > 50% efficiency gain compared with conventional lab testing (typically 30–60 minutes) [18,25]. These features further enable the practical implementation of dynamic bedside LqSOFA monitoring, offering operational solutions for primary care hospitals.

However, our study revealed considerable variability in LqSOFA sensitivity (range: 0.31–0.64), which remains a central challenge for clinical implementation. While its high specificity effectively rules out low-risk patients, its suboptimal sensitivity limits early identification of high-risk patients. This phenomenon is attributable primarily to inherent limitations in the assessment framework, where significant heterogeneity exists in threshold selection for both qSOFA scores and lactate levels across current protocols. Certain studies define positivity using qSOFA ≥2 points combined with lactate ≥2 mmol/L [12,21,37], consequently excluding patients with occult shock (qSOFA = 1 point but elevated lactate) from the high-risk cohort. This cohort of patients without hypotension (SBP > 100 mmHg) that exhibited tissue hypoperfusion (lactate ≥2 mmol/L) constituted 21.3% (95% CI: 18.7–24.1) of the ED sepsis population. These patients demonstrated significantly elevated SOFA scores, indicating increased risks of organ dysfunction and shock progression [32]. Similarly, Hwang et al. reported that 26.6% of sepsis patients initially presented with occult shock (lactate ≥4 mmol/L with normotension), with 72.4% progressing to manifest shock within 72 hours. Despite meeting single-point lactate thresholds, these patients experienced progressive deterioration, resulting in mortality rates as high as 27.4% [45]. Notably, a key contributor to LqSOFA's suboptimal sensitivity may be its reliance on single-point lactate measurements, which fail to identify patients with progressively deteriorating conditions. Daga et al.'s prospective study demonstrated that dynamic lactate clearance rate, not isolated lactate levels, serves as a critical prognostic indicator, with delayed clearance (<10%/hour) significantly increasing the mortality risk (adjusted OR 4.2, 95% CI: 2.7–6.5) [18]. This underscores the potential for incorporating serial lactate measurements into prognostic models to increase sensitivity. Given the dual limitations of single-point lactate measurement and inconsistent threshold standards, future research should (1) validate the real-world survival benefits of dynamic lactate monitoring (e.g., serial measurements every 2–4 hours) through multicenter prospective studies and (2) establish evidence-based precision cutoffs, ultimately developing context-specific risk identification frameworks for sepsis across diverse healthcare environments.

## 5. Discussion of limitations

This study has several limitations that require future research attention: (1) its exclusive focus on ED/ICU settings (100%) lacks data from critical LMIC contexts such as general wards and prehospital environments; (2) the absence of pathogen stratification (viral, bacterial, fungal) potentially underestimates biomarker differences (e.g., lower PCT in viral sepsis), impacting LqSOFA performance; (3) reliance on single biomarker measurements overlooks the value of dynamic indicators (e.g., lactate clearance) for prognosis and optimizing monitoring sensitivity; (4) significant heterogeneity in LqSOFA definitions (variable lactate cutoffs ≥2- ≥ 4/L, inconsistent logic) combined with insufficient data prevents analysis of their impact, limiting generalizability; (5) limited LMIC validation scope (7 studies, primarily Asian) necessitates broader assessment in diverse regions (e.g., Sub-Saharan Africa, Latin America) to gauge robustness across contexts; and (6) potential biases may arise from the predominance of retrospective studies (19/30), with the risk of selection bias/ incomplete data, while Deeks' funnel plot asymmetry for SOFA suggests possible publication bias that may influence the results.

## 6. Conclusion

The SOFA score remains the optimal predictor of sepsis mortality risk, whereas the qSOFA score demonstrates suboptimal overall predictive ability. LqSOFA achieves comparable accuracy to SOFA by synergistically combining the advantages of lactate and qSOFA with high specificity, which is particularly valuable for rapid risk exclusion in resource-limited settings (ED/LMICs). Future studies should validate LqSOFA across diverse clinical settings and underrepresented LMIC regions and explore the integration of dynamic lactate clearance metrics.

## Supporting information

**S1 Fig. Distribution of Predictor Combinations in Included Studies.** (a) SOFA vs. PCT; (b) SOFA vs. Lactate; (c) SOFA vs. qSOFA; (d) SOFA vs. LqSOFA; (f) LqSOFA vs. qSOFA. Caption: Venn diagrams quantify study overlap between predictors: Blue circles represent studies reporting SOFA data (Panel a: n = 8), green circles represent comparator metrics (Panel a: PCT n = 2), intersection values indicate studies with complete paired data (confusion matrices + AUROC/95% CI; Panel a: n = 10), and yellow circles with arrows represent studies with only AUROC/95% CI pairs (Panel a: n = 2). Analytical approach: 1) Metrics for individual predictors use all studies in their colored circles (e.g., SOFA specificity: 8 + 10 = 18 studies); 2) AUROC comparisons combine intersection and yellow-circle studies (e.g., SOFA vs. PCT: 10 + 2 = 12 studies).
(TIF)

**S2 Fig. Risk of bias and applicability concerns graph of included studies.**
(TIF)

**S3 Fig. Risk of bias and applicability concerns graph of included studies.**
(TIF)

**S4 Fig. Risk of bias and applicability concerns graph of included studies.**
(TIF)

**S5 Fig. Forest plot of pooled sensitivity and specificity.** (a) PCT; (b) Lactate; (c) qSOFA; (d)Lqsofa; (f)Sofa.
(TIF)

**S6 Fig. Deek's funnel plot for publication bias.** (a) PCT; (b) Lactate; (c) qSOFA; (d)Lqsofa; (f)Sofa.
(TIF)

**S7 Fig. The results of univariable meta – regression and subgroup analyses.** (a) PCT; (b) Lactate; (c) qSOFA; (d) Lqsofa; (f) Sofa.
(TIF)

**S1 Table. Search Strategies.**
(DOCX)

**S2 Table. Subgroup Analyses of Pooled Diagnostic Performance of PCT in Predicting Sepsis Patient Mortality.**
(DOCX)

**S3 Table. Subgroup Analyses of Pooled Diagnostic Performance of Lactate in Predicting Sepsis Patient Mortality.**
(DOCX)

**S4 Table. Subgroup Analyses of Pooled Diagnostic Performance of qSOFA in Predicting Sepsis Patient Mortality.**
(DOCX)

**S5 Table. Subgroup Analyses of Pooled Diagnostic Performance of SOFA in Predicting Sepsis Patient Mortality.**
(DOCX)

**S1 File. PRISMA 2009 checklist.**
(DOCX)

## Author contributions

**Conceptualization:** Jinmei Lu, Zhouzhou Dong, Longqiang Ye, Zaixing Zheng.

**Data curation:** Jinmei Lu, Yi Gao, Zaixing Zheng.

**Funding acquisition:** Zaixing Zheng.

**Methodology:** Jinmei Lu, Zaixing Zheng.

**Writing – original draft:** Jinmei Lu, Longqiang Ye, Zaixing Zheng.

**Writing – review & editing:** Jinmei Lu, Zaixing Zheng.

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
