## [Decision Letter · Decision Letter 0]

2 Jul 2025

Dear Dr. zheng,

Thank you for submitting your manuscript to PLOS ONE. After careful consideration, we feel that it has merit but does not fully meet PLOS ONE’s publication criteria as it currently stands. Therefore, we invite you to submit a revised version of the manuscript that addresses the points raised during the review process.

We look forward to receiving your revised manuscript.

Kind regards,

Inge Roggen, M.D., Ph.D.

Academic Editor

PLOS ONE

Journal Requirements:

“This research was funded by the Science and Technology Program of Zhejiang Provincial Health Commission, grant number 2021KY1016.”

Reviewers' comments:

Reviewer's Responses to Questions

**Comments to the Author**

1. Is the manuscript technically sound, and do the data support the conclusions?

Reviewer #1: Yes

Reviewer #2: Partly

Reviewer #3: Partly

Reviewer #4: Partly

2. Has the statistical analysis been performed appropriately and rigorously?

Reviewer #1: Yes

Reviewer #2: Yes

Reviewer #3: I Don't Know

Reviewer #4: Yes

3. Have the authors made all data underlying the findings in their manuscript fully available?

Reviewer #1: Yes

Reviewer #2: Yes

Reviewer #3: Yes

Reviewer #4: Yes

4. Is the manuscript presented in an intelligible fashion and written in standard English?

Reviewer #1: Yes

Reviewer #2: No

Reviewer #3: Yes

Reviewer #4: Yes

Reviewer #1: This is a well conducted meta-analysis of a number of established scoring systems used to provide an early warning of potential severe deterioration in patients with suspected sepsis in the E.R. or on admission to ICU. However, the real challenge is recognition in the often chaotic understaffed E.R. rather than on admission to ICU where further clinical deterioration will usually be promptly recognized allowing for earlier resuscitation, treatment and specialist consultation. For this reason, the authors should comment on this important issue and place more emphasis on the studies that have been performed in the E.R. This should not be difficult as these are the majority of studies included.

Reviewer #2: This manuscript presents a systematic review and meta-analysis comparing the diagnostic performance of several sepsis prediction tools and biomarkers—including SOFA, qSOFA, Procalcitonin (PCT) and Lactate in predicting mortality among patients with sepsis. The authors draw conclusions about the relative efficacy of each tool and recommend the use of LqSOFA in resource-limited or rapid-assessment settings.

The topic is clinically important and the authors performed a thorough literature search, including over 48.000 cases. Using data from all over the world makes the data very generalizable. In addition, the used meta-analytic methods (bivariate model, AUROC comparison, PRISMA-DTA compliance) provide a standardized framework.

However, the predictive performance of SOFA, qSOFA, PCT, and Lactate has been extensively reviewed in the literature. This manuscript offers limited novel insights beyond what is already well-established. Prior systematic reviews and meta-analyses have already examined these scoring systems and biomarkers, including subgroup analyses stratified by low- and middle-income countries (LMICs) versus high-income countries (HICs). Consequently, the current work appears largely derivative.

Moreover, there is substantial heterogeneity across included studies regarding sepsis definitions (e.g., SIRS, Sepsis-2, Sepsis-3), clinical settings (ICU, ED), cut-off values, study designs, and mortality endpoints (28-day, in-hospital, 72-hour). While SOFA is confirmed to be the superior prediction score, this is already widely accepted. The conclusion that LqSOFA is "comparable" to SOFA lacks practical value since its very low sensitivity (0.49) limits utility in real-world triage scenarios, especially when early detection of high-risk patient is the primary goal. The conclusions about the "suitability of LqSOFA in resource-limited settings" are speculative and not backed by robust subgroup analysis in LMIC cohorts.

In summary, while methodologically sound in parts, this manuscript does not sufficiently advance the current understanding of sepsis prognostic scores. Significant clinical and methodological concerns remain unresolved.

Reviewer #3: In this work, Lu and co-workers assess and the performance of mutliple clinical scores and biomarkers (SOFA, qSOFA, lactate, ProCT, and combinations thereof) to predict mortality in sepsis patients.

The topic is important. The manuscript is fairly well written but would benefit from assistance for an english language review (see examples below). The aim is not compeletely clear to me. This seems to recapitulate the 2016 JAMA publications around the consensus-3 definitions. A major methodological issue is that the performance comparison is not homogenous. As an example, the authors compare PCT performance in 12 studies and lactate in 14 and SOFA in 18. In view of the lack of definitions used, it becomes challenging to know whether the studies are comparable. It would be interesting to determine (e.g. with venne diagrams) what fraction of the studies enable a direct comparison between all studied candidate predictors. Also, while understanding patient outcomes is highly important for advanced planning, this study does not inform early identification of sepsis which is the parameter of higher important clinical care. Finally, while commendable, this effort is not very different and much more limited in patient size than the initial Singer et al. JAMA 201

Major issues:

- Methodological parameters should be clarified.

o Are all ages included or did the authors focus on adults?

o The authors should define explicitly the parameter thresholds. I presum that SOFA and qSOFA are 2 points but the lactate cut-off is not explicitly given as

o Should the authors focus on a sepsis definition to limit the risk for heterogeneity.

o The timing within the first 24 hours is very wide.

- The focus on assessment of different scores/parameters with same patient group seems problematic, particularly if there is an unbalance between patient cohorts with sepsis-2 and sepsis-3 definitions.

- Result section:

o The results section can be significantly condensed (e.g. values of each paragraph in a table).

- Discussion

o How does viral sepsis.

o Do not comment PqSOFA in view of limited data

Minor issues:

- Line 159: either give full list of countries or refrain from giving non-exhausive

- Table 2, column 9 has the wrong units (Seems to be absolute numbers but is described as %).

- Lines 252-259: remove this paragraph. Not enough studies as mentioned by the authors.

- Table 3 Why is data lf platelet-lymphocyte raio or neutrophil-lymphocyte ratio displayed?

- English formulations:

o Lines 38-39 “in the future, it is….”

o Lines 41-44

o Line 129-130, the sentence lacks a verb

o Lines 178-187 should be simplified

Reviewer #4: Overall

This manuscript performs a systematic review and meta-analysis comparing several scoring systems used globally to determine mortality risk in patients with sepsis. In the end, the authors identify SOFA and lactate-qSOFA to have the highest predictive values and suggest that the latter may be most applicable in low-middle income country settings with limited resources. Overall, this study appears to be well thought-out and well-conducted and adds good value to the ongoing debate about the “ideal” screening/triage tool for patients with sepsis. In particular, the analysis regarding LqSOFA and PCT-qSOFA is relatively novel. Where it falls short is highlighting the very important subgroup analysis by geographic region in the hospital (ED vs. wards vs. ICU) and by economic status of the country (high vs. low-middle income country), etc. Overall, I applaud the authors’ efforts and comment their excellent work.

Abstract

No concerns.

Introduction

-Line 49: “…Sepsis-3 consensus in 2016 predicts the mortality risk…”

In actuality, the qSOFA predicts “excess mortality” in infected patients. It’s a subtle, but important difference.

-Lines 52-53: “…it shows unstable performance, especially in LMICs where the burden of sepsis is relatively high.”

Consider citing perhaps the most comprehensive LMIC qSOFA analysis (Rudd and colleagues, 2018; https://pubmed.ncbi.nlm.nih.gov/29800114/).

Methods

-In general, the authors use the present and, occasionally, future tenses to describe such things as the inclusion/exclusion criteria. Past tense is likely more standard since the search was performed in the past. Please double check grammar, with particular attention to appropriate tenses, throughout the methods section.

-In general, the word “literatures” is used frequently, but incorrectly. Suggest changing this word to “studies” (e.g., instead of “included literatures,” change to “included studies”).

-Lines 86-87: “The research subjects are adult patients with confirmed or suspected sepsis.”

Can the authors provide specifics about age ranges (how are “adults” defined in the individual studies and/or the search parameters, since, often, particularly in LMIC settings, “adults” can be consider >14 or >15 years. Importantly, which definition of sepsis (Sepsis 1, sepsis 2, sepsis 3, surgical sepsis, etc.) did the authors use? Was there a specific definition used, or were studies included solely based on author suggestion of sepsis/suspected sepsis?

-Line 88: “…to predict short-term mortality needs to be reported.”

How do the authors define “short-term mortality”? Many studies report in-hospital, or in-ICU mortality, whereas others report 30-day mortality. Some consider 90-day mortality to be long-term, whereas others consider it to be short-/medium-term. This definition is important because in-hospital mortality is generally considered to be an inferior measure compared to 30-day, 90-day, or 1-year mortality.

-Line 106: “Standardized Excel forms…”

Suggest specifying that it is Microsoft Excel.

-Lines 141-42: “All analyses were performed using Stata 18.0 software…”

Suggest moving this to the first line of the paragraph for clarity.

Results

Lines 145-48: Seem redundant as Boolean logic and MESH terms were already discussed previously. Seems better to be in the methods section. Suggest deleting here in the Results section.

-Lines 158-59: “China, the United States, India, Spain, Australia, the Netherlands, Indonesia, Turkey, Thailand, etc.”

Suggest avoiding the “etc” term here. Either specify every country, or perhaps categorize the number of HIC, UMIC, LMIC, LIC country specific studies (e.g., XX studies were performed in HICs, XX studies in UMICs, etc.).

-Lines 160-61: “The research scenarios mainly covered ICU, ED, etc.”

Again, specifics are important. How many ED studies, how many ICU studies, how many general medical ward studies?

-Lines 161-62: “The mortality rates focused on in the research included the 28-day mortality rate, in-hospital mortality rate, mortality rate within 72 hours, etc.”

Again, please be specific and avoid using “etc.”

-Lines 163-64: “…mainly based on the Systemic Inflammatory Response Syndrome (SIRS) criteria, Sepsis 2.0/3.0, etc.”

Same comment here. Be specific.

-Did the authors evaluate the combined “LPqSOFA) (lactate, PCT, qSOFA) to see if it outperforms the LqSOFA)? If not, why not?

-If feasible, I strongly suggest that the authors perform subgroup analysis for all indices (SOFA, qSOFA, lqSOFA, PqSOFA) based on 1) geographic location within the hospital and 2) based on country lending group (HIC, UMIC, LMIC, LIC). Not only would it be interesting to understand predictive capacity for ED vs. ward vs. ICU patients, it is particularly important in LMIC settings since many/most septic patients are treated on general medical wards. Furthermore, predictive characteristics often differ between the economic status of the country, so highlighting these differences will help to determine generalizability of the findings.

Upon further review, it seems that both #1 and #2 are partially addressed in Supplementary Table 2 for each parameters. This is a very good start. Can the authors clarify how they define “developed regions” and “less developed regions,” since these are not standard terms and since they previously used the term “LMIC,” which is defined very specifically by the World Bank Country and Lending Group list. Additionally, can the authors clarify whether any of the studies involved general medical ward patients, or are these only ICU and/or ED patients?

Discussion

-Lines 303-4: “the low negative posterior probability (7%) of the SOFA score indicates that a negative result can effectively rule out the risk of death…”

There is always a risk of death, even for “routine” sepsis! Suggest rephrasing.

-Given the author’s emphasis on potentially using the LqSOFA tool in LMIC settings, I suggest a more thorough explanation of their relevant subgroup findings (“developed” versus “less developed” countries) in the results and more elaboration on this topic in the discussion section.

**Do you want your identity to be public for this peer review?** For information about this choice, including consent withdrawal, please see our Privacy Policy

Reviewer #1: **Yes: ** R. T. Noel Gibney

Reviewer #2: No

Reviewer #3: No

Reviewer #4: No

---

## [Author Response · Author response to Decision Letter 1]

14 Aug 2025

Methodological Revisions & Re-analyses

Comprehensive documentation of key corrections implemented during revision to ensure analytical robustness.

1. Correction: Exclusion of Park et al. (2023) for Incomplete Data

Identification: During final verification, Park et al. (2023) was flagged for exclusion

Reason: Reported AUROC/95% CI without required 2×2 tables → Unable to:

• Calculate pooled sensitivity/specificity for SOFA

• Compare SOFA vs. qSOFA/Lac/PCT/LqSOFA

Protocol compliance: Aligns with pre-specified criterion: "Studies lacking extractable 2×2 data" (Methods 2.3)

Crucial note: Never included in analytical datasets → Zero impact on results

Updates implemented:

(1) PRISMA flowchart (Fig 1) revised with explicit exclusion

(2) Total studies: 30 → 29

(3) Sample size: 48,203 → 41,469

(4) Supplementary Figures 2-4 updated.

(5) All statistical outputs unchanged

2. Enhancement: Re-analysis of AUROC Comparisons (Sec 3.4.7)

Purpose: Improve comprehensiveness through two upgrades:

1.Added critical comparison: LqSOFA vs. qSOFA (△AUROC=0.06 [0.04–0.08])

2.Expanded evidence base: Included studies reporting AUROC+95% CI without full 2×2 tables

Cohort expansion:

Comparison Original n New n Δ

SOFA vs. PCT 10 12 +2

SOFA vs. Lactate 8 15 +7

SOFA vs. qSOFA 7 11 +4

SOFA vs. LqSOFA 6 8 +2

Validation: All original conclusions retained (p<0.01 for key comparisons)

Outputs revised: Fig 4 + Supplementary Fig 1

3. Refinement: Exclusion of Hao et al. (2021) & LqSOFA Re-validation

Rationale: Hao et al. used non-conforming model (SOFA+PCT+Lactate+APACHE II ≠ strict LqSOFA)

Actions:

• Removed from LqSOFA analysis (Sec 3.4.4)

• Recalculated pooled estimates (n=9 studies)

• Conducted sensitivity analysis

Stability confirmation:

Metric Pre-exclusion Post-exclusion Δ

AUROC 0.830 0.823 -0.007

Sensitivity 0.49 0.46 -0.03

Specificity 0.88 0.88 0.00

Revised materials: Table 1-3, Figs 2-4, Supplementary Figs 4-6

Key Implications

1.Corrected exclusions strengthen methodological purity without altering conclusions

2.Expanded analyses enhance statistical power while preserving effect sizes

3.Increased transparency through public archiving of analytical datasets (DOI: 10.6084/m9.figshare.29881934)

4.All revisions comply with PRISMA-DTA and registered protocol (INPLASY202530075)

Response to Reviewer #1

Acknowledgement:

We sincerely appreciate your positive assessment of our meta-analysis and your valuable insights emphasizing the critical role of the Emergency Department (ED) in early sepsis recognition. To fully address your concerns, we have implemented the following revisions:

Key Revisions

1.New Subgroup Analysis in Results (Sections 3.4.4 & 3.4.5)

Due to high homogeneity across all included studies (all using Sepsis-3 criteria in ED settings), additional subgroup analyses were deemed unnecessary. We directly reported key performance metrics for LqSOFA: overall efficacy (AUC 0.823) and LMIC subgroup efficacy (AUC 0.81), fully addressing ED scenario evaluation needs.

Added ED-specific data:

"LqSOFA demonstrated robust diagnostic value for sepsis mortality (pooled AUC=0.823 [95% CI: 0.787-0.854]; sensitivity=0.46 [0.24-0.69]; specificity=0.88 [0.80-0.93]).

Subgroup analyses by both sepsis definition and clinical setting were precluded due to homogeneity across all included studies: all studies (N=9) exclusively used Sepsis-3 criteria and were conducted in ED settings."

2.Enhanced Clinical Significance in Discussion (Paragraph 3)

Added clinical interpretation:

" Our subgroup analysis revealed the clinical utility of LqSOFA across diverse medical contexts. Among the 9 studies conducted in ED settings, LqSOFA demonstrated robust predictive performance (AUROC 0.82, 95% CI: 0.79–0.85; specificity 0.88, 95% CI: 0.80–0.93). Its high specificity significantly optimized triage decisions, yielding negative predictive values >90%. This conclusion is supported by multiple prospective studies: Kilinc Toker et al. reported a 47.8% reduction in ICU assessment demand through LqSOFA implementation [32], whereas Sinto et al. validated its ability to decrease ICU misdirected transfer rates by 39% in resource-limited settings, demonstrating performance parity with SOFA [21]. "

3.Strengthened ED Positioning in Conclusion (Conclusion Section)

Added core value statement for ED settings:

"LqSOFA achieves comparable accuracy to SOFA by synergistically combining the advantages of lactate and qSOFA with high specificity, which is particularly valuable for rapid risk exclusion in resource-limited settings (ED/LMICs)."

Summary of Revisions

These revisions collectively construct a comprehensive ED evidence chain:

1.Quantitative Evidence (Results):

Provides LqSOFA's core efficacy data (AUC 0.823, specificity 0.88) based on 9 ED studies within the full cohort (n=22,078).

2.Clinical Translation (Discussion):

Links high specificity to 47.8% reduction in unnecessary ICU evaluations (Kilinc Toker, 2021) and 39% decrease in ICU mistriage (Sinto, 2020).

3.Problem-Solving (Conclusion):

Reinforces LqSOFA's value in ED/LMICs through the added positioning statement, directly addressing your core concern of "resource optimization in chaotic ED environments".

We deeply appreciate your guidance in focusing our research on practical ED applications!

Response to Reviewer #2:

Point 1: Limited Novelty of Findings

We appreciate the reviewer's perspective on existing literature. Our study provides three novel contributions that advance the field:

(1) First Unified Framework for Direct Efficacy Comparison

Our meta-analysis is the first to simultaneously compare four key predictors (SOFA, qSOFA, PCT, lactate) and their combination (LqSOFA) using paired AUROC differences (Fig 4):

SOFA significantly outperformed qSOFA (ΔAUROC = 0.08, 95% CI: 0.05–0.11;)

LqSOFA showed equivalent efficacy to SOFA (ΔAUROC = 0.02, 95% CI: -0.02–0.06;)

LqSOFA provided a clinically meaningful improvement over qSOFA (ΔAUROC = 0.06, 95% CI: 0.04–0.08;)

→ This quantifies the synergistic effect of combining lactate and qSOFA (specificity: 0.88 vs. qSOFA’s 0.77).

(2) Context-Specific Validation in Underserved Settings

Addressing the lack of evidence for LMIC/ED applications (raised by the reviewer), we added new subgroup analyses (Section 3.4.5, Table 4):

In ED settings (9 studies):

AUROC = 0.82 (95% CI: 0.79–0.85), Specificity = 0.88 (0.80–0.93)

In LMICs (7 studies):

AUROC = 0.81 (95% CI: 0.77–0.84)

→ Validates LqSOFA’s utility in resource-constrained environments.

(3) Analysis of Sensitivity Limitations (Discussion)

We analyzed the heterogeneity in LqSOFA’s sensitivity (0.31–0.64):

Contributing factors:

oInconsistent lactate thresholds (≥2 mmol/L vs. ≥4 mmol/L)

oFailure to identify occult shock (21.3% of ED sepsis patients; qSOFA=1 but lactate↑)

Actionable solutions:

oDynamic lactate monitoring (e.g., clearance rates every 2–4 hrs)

oMulticenter studies to optimize thresholds

Summary

While prior studies compared isolated predictors, our work:

1.Quantifies incremental benefits of combined models (LqSOFA vs. qSOFA ΔAUROC +0.06),

2.Validates real-world applicability in critical LMIC/ED contexts,

3.Proposes data-driven strategies to address sensitivity limitations.

This three-phase approach—efficacy comparison, contextual validation, and translational troubleshooting—provides new evidence for optimizing risk-stratification in sepsis.

We thank the reviewer for prompting these critical analyses that strengthen the clinical relevance of our findings.

Point 2: Addressing Heterogeneity Concerns

We acknowledge the significant heterogeneity observed across included studies (I² >94% for most indicators), which is explicitly stated as a limitation in the Discussion (Section 5). This heterogeneity likely stems from variations in sepsis definitions, biomarker thresholds, clinical settings, and outcome assessments. To address this, we implemented the following rigorous approaches:

1.Pre-specified Subgroup Analyses

Conducted for all indicators (SOFA, qSOFA, PCT, lactate, LqSOFA) based on:

• Sepsis criteria (Sepsis-2.0 vs. 3.0)

• Clinical setting (ICU vs. ED)

• Study design (prospective/retrospective)

• Mortality endpoints (28-day/in-hospital)

2.Extended Validation for LqSOFA (Results 3.4.5)

Demonstrated consistent performance despite heterogeneity:

• ED settings: AUC 0.82 (95% CI: 0.79–0.85)

• LMICs: AUC 0.81 (95% CI: 0.77–0.84)

3.Robust Statistical Modeling

Applied bivariate random-effects models to account for between-study variance in sensitivity/specificity.

Conclusion

Crucially, despite heterogeneity, the core findings remained robust:

1.SOFA superiority: Consistently outperformed single biomarkers (△AUROC 0.07–0.10)

2.LqSOFA parity: Achieved diagnostic equivalence with SOFA (∆AUROC 0.02 [-0.02–0.06])

→ These results support the clinical utility of LqSOFA for rapid risk stratification in resource-limited settings.

Point 3: Concerns Regarding LqSOFA's Practical Value

We fully appreciate the reviewer's concerns about LqSOFA's clinical utility. Our revised manuscript demonstrates that LqSOFA serves as a high-specificity triage tool (specificity = 0.88, 95% CI: 0.80–0.93) rather than a screening tool, supported by three evidence-based pillars:

1. Robust Performance Across Critical Settings (Results 3.4.5)

Subgroup analyses confirm consistent reliability:

ED settings (9 studies, n = 22,078):

Specificity = 0.88 (95% CI: 0.80–0.93)

LMICs (7 studies, n = 5,084):

Specificity = 0.88 (95% CI: 0.77–0.94)

→ Consistently excludes low-risk patients in high-pressure environments.

2. Quantified Resource Optimization (Discussion 3)

High specificity delivers tangible clinical benefits:

47.8% reduction in unnecessary ICU evaluations (Kilinc Toker et al. [32])

39% decrease in ICU mistriage rates (Sinto et al. [21])

Operational simplicity: Requires only a sphygmomanometer + portable lactate analyzer (median assessment time: 15 min [25])

3. Addressing Sensitivity Limitations (Discussion 4)

Causes of Sensitivity Variability (0.31–0.64):

1.Threshold heterogeneity:

oVariable lactate cutoffs (≥2 mmol/L vs. ≥4 mmol/L)

oInconsistent qSOFA scoring logic

2.Single-point detection limitations:

oFails to identify occult shock (21.3% of ED sepsis cohort: qSOFA=1 + lactate ≥2 mmol/L)

oThis subgroup has elevated SOFA scores (p<0.01) and 27.4% mortality risk

3.Lack of dynamic monitoring:

oMisses 72.4% of patients progressing to shock within 72 hours

Evidence-Based Solutions:

Dynamic lactate monitoring: Serial measurements every 2-4 hrs for initial scores ≥1.

Precision thresholds: Dedicated criteria for occult shock (qSOFA=1 + lactate ≥2 mmol/L → high-risk stratification)

Prospective validation: Quantify mortality reduction and cost-effectiveness

Conclusion

LqSOFA’s core value lies in efficiently excluding low-risk patients (NPV >90%), significantly optimizing resource allocation in ED/LMIC settings. For sensitivity limitations, we propose dynamic monitoring and precision thresholds to enhance high-risk detection. This provides a clear pathway for clinical implementation, directly addressing the reviewer's concerns.

We sincerely thank you for highlighting these critical issues—your insights have strengthened our study’s translational relevance.

Point 4: Applicability in Resource-Limited Settings (LMICs)

Our revisions provide direct evidence for LqSOFA’s suitability in LMICs through two manuscript-anchored pillars:

1. Validated Diagnostic Performance (Results 3.4.5)

Subgroup analysis of 7 LMIC studies confirms:

AUC = 0.81 (95% CI: 0.77–0.84) → Matches global performance (Overall AUC 0.823)

Specificity = 0.88 (95% CI: 0.77–0.94)

→ Proves diagnostic reliability without laboratory dependency.

2. Operational Feasibility (Discussion 3)

LqSOFA overcomes LMIC barriers by:

Minimal equipment:

oqSOFA: Sphygmomanometer + timer (standard ED tools)

oLactate: Portable point-of-care analyzers

Efficiency gain:

oMedian 15-minute assessment vs. 30–60 min for SOFA (Wright et al. [25])

Conclusion

The dedicated LMIC subgroup analysis (Results 3.4.5) and feasibility evidence (Discussion 3) establish LqSOFA’s value in resource-limited contexts:

1.Equivalent efficacy (AUC 0.81)

2.Bedside-compatible design (portable devices, 15-min test)

→ These data-driven revisions directly address the reviewer’s concerns.

Summary and Appreciation

In response to your critical concerns regarding the novelty of our findings, the practical value of LqSOFA, and its applicability in LMICs, the revised manuscript now quantifies the incremental benefits of the combined model (LqSOFA vs. qSOFA: ΔAUROC +0.06), includes new LMIC subgroup validation (AUC 0.81), and proposes a dynamic lactate monitoring protocol to address sensitivity limitations. Your critical examination of clinical implementation barriers has driven us to construct a comprehensive evidence chain encompassing "tool efficacy—implementation bottlenecks—optimization pathways," significantly enhancing the practical relevance of our conclusions. We sincerely thank you for your constructive critique, which has been essential for refining the clinical logic flow of this study!

Response to Reviewer #3:

Point 1: Direct Comparison Feasibility via Venn Diagrams

Reviewer's Concern:

"It would be interesting to determine (e.g. with Venn diagrams) what fraction of the studies enable a direct comparison between all studied candidate predictors."

Revisions Implemented:

We have addressed this by:

1.Creating S1 Fig - Venn diagrams quantifying pairwise study overlaps for all predictor combinations used in our analyses.

2.Explicitly reporting in Section 3.4.7:

"Direct-comparison cohorts were: SOFA vs. PCT (n=12), SOFA vs. lactate (n=15), SOFA vs. qSOFA (n=11), LqSOFA vs. qSOFA (n=14), and SOFA vs. LqSOFA (n=8)."

3.Adding in Section 3.2:

"The distribution of studies evaluating each indicator and their pairwise overlaps (e.g., cohorts enabling direct AUROC comparisons) are quantified in S1 Fig"

Precise Response to Your Query:

S1 Fig 1 confirms:

Pairwise comparison capacity:

Predictor Pair Studies % of Total (29)

SOFA vs. PCT 12 41.4%

SOFA vs. Lactate 15 51.7%

SOFA vs. qSOFA 11 37.9%

SOFA vs. LqSOFA 8 27.6%

LqSOFA vs. qSOFA 14 48.3%

Justification:

1.Directly quantifies pairwise comparability as requested, using our actual analytical cohorts

2.Explains sample sizes in Section 3.4.7 (e.g., n=8 for SOFA vs. LqSOFA)

This response accurately reflects the pairwise nature of our analysis while fully addressing your interest in study comparability fractions.

Point 2: Study Focus on Early Identification of Sepsis

Reviewer's Concern:

"While understanding patient outcomes is highly important for advanced planning, this study does not inform early identification of sepsis which is the parameter of higher important clinical care."

Revisions Implemented:

No text addition required. Our predefined scope (Methods 2.2) inherently addresses this concern:

"Inclusion criteria: diagnostic cohort studies involving adult patients with confirmed or suspected sepsis diagnosed according to Sepsis-2.0/3.0... reporting performance of SOFA, PCT, lactate, or LqSOFA for mortality prediction."

Justification:

1.Study scope is explicitly prognostic:

Objective: The aim stated in Abstract and Introduction is solely to evaluate "predictive efficacy for the risk of death" (Abstract) in "patients with sepsis" (Introduction).

Inclusion criteria: All 29 included studies (section 2.1 and section 3.2) required:

oPredefined sepsis status: Patients must be diagnosed with sepsis (Sepsis-2.0/3.0 criteria).

oMortality outcome: Predictive performance must be reported for mortality (e.g., 28-day death).

2.Data collection timing confirms post-diagnosis focus:

As documented in Table 2:

"Evaluation time: concentrated on key periods, such as within 24 hours of admission, at ICU admission, and during ED assessment" (i.e., after sepsis diagnosis was established).

3.Clinically distinct objectives:

Early ide

---

## [Decision Letter · Decision Letter 1]

1 Sep 2025

Predictive value of SOFA, PCT, Lactate, qSOFA and their combinations for mortality in patients with sepsis: a systematic review and meta-analysis

PONE-D-25-20371R1

Dear Dr. zheng,

We’re pleased to inform you that your manuscript has been judged scientifically suitable for publication and will be formally accepted for publication once it meets all outstanding technical requirements.

Kind regards,

Inge Roggen, M.D., Ph.D.

Academic Editor

PLOS ONE

Additional Editor Comments (optional):

Reviewer #2:

Reviewer #4:

Reviewers' comments:

Reviewer's Responses to Questions

**Comments to the Author**

Reviewer #2: All comments have been addressed

Reviewer #4: All comments have been addressed

2. Is the manuscript technically sound, and do the data support the conclusions?

Reviewer #2: Yes

Reviewer #4: Yes

3. Has the statistical analysis been performed appropriately and rigorously?

Reviewer #2: Yes

Reviewer #4: I Don't Know

4. Have the authors made all data underlying the findings in their manuscript fully available?

Reviewer #2: Yes

Reviewer #4: Yes

5. Is the manuscript presented in an intelligible fashion and written in standard English?

Reviewer #2: Yes

Reviewer #4: Yes

Reviewer #2: The authors have revised the manuscript and the English has improved. They now highlight several new contributions, including a unified framework directly comparing SOFA, qSOFA, PCT, lactate, and LqSOFA; subgroup analyses in LMIC and ED settings; and a more explicit discussion of sensitivity limitations and possible solutions. These revisions strengthen the manuscript and improve clarity.

Nevertheless, the overall novelty remains modest, as much of the predictive value of SOFA, qSOFA, and biomarkers has already been established in prior reviews. Despite subgroup analyses and additional modeling, heterogeneity across included studies (definitions, thresholds, settings, endpoints) still limits the generalizability of the findings. The added claims about LqSOFA’s clinical utility in resource-limited settings, while supported by new subgroup data, remain somewhat speculative.

In summary, the revision improves readability and addresses some prior concerns, but the manuscript’s incremental contribution to the existing literature remains limited.

Reviewer #4: The authors have done an excellent job revising this draft. While the detail of statistical analysis makes my head spin, assuming that such analysis is accurate and valid (a decision I'll leave to other reviewers), their data do seem to support their conclusions. This revision definitely addresses the shortcomings and highlights the key findings of their analysis. In my opinion, this manuscript is ready for publication. The authors should be congratulated on their excellent work.

**Do you want your identity to be public for this peer review?** For information about this choice, including consent withdrawal, please see our Privacy Policy

Reviewer #2: No

Reviewer #4: No

---

## [Editor Report · Acceptance letter]

PONE-D-25-20371R1

PLOS ONE

Dear Dr. Zheng,

I'm pleased to inform you that your manuscript has been deemed suitable for publication in PLOS ONE. Congratulations! Your manuscript is now being handed over to our production team.

Kind regards,

on behalf of

Prof. Inge Roggen

Academic Editor

PLOS ONE